# Main Factors Affecting Animal Welfare and Meat Quality in Lambs for Slaughter in Chile

**DOI:** 10.3390/ani8100165

**Published:** 2018-09-27

**Authors:** Carmen Gallo, Juan Tarumán, Cristian Larrondo

**Affiliations:** 1Animal Science Institute, Faculty of Veterinary Science, OIE Collaborating Centre for Animal Welfare and Livestock Production Systems—Chile, Universidad Austral de Chile, Valdivia 5090000, Chile; 2Faculty of Veterinary Science, Universidad Austral de Chile, Valdivia 5090000, Chile; juantaruman@gmail.com (J.T.); cristian.larrondoc@gmail.com (C.L.)

**Keywords:** animal welfare, meat quality, lambs, transport, preslaughter operations

## Abstract

**Simple Summary:**

Consumers have become increasingly demanding about the quality of the meat they eat; they want healthy, natural, and palatable products, but are also concerned about animal welfare during production, transport, and slaughter. The way we handle animals does not only affect animal welfare, but also the quality of the meat produced. In this study the main factors affecting the welfare of lambs for slaughter in Chile are reviewed and the implications on meat quality shown. Factors like long distance transport due to difficult geographical reasons, bad roads, and lack of adequate local slaughterhouses appear to have a major impact on the welfare of lambs and also in reducing the quantity and quality of meat produced. Some deficiencies in terms of the handling of the lambs by untrained stockmen and drivers appear, and also inadequate infrastructure for transport in terms of vehicles and loading/unloading ramps. The problems observed are common for other South American countries and should be addressed firstly by educating and training all people involved in the lamb meat chain regarding animal welfare. In Chile there is legislation, ongoing since 2013, regarding the protection of animals during production, transport, and slaughter, including compulsory training of the people involved, which should improve animal welfare and meat quality.

**Abstract:**

Consumers have become increasingly demanding about the quality of products of animal origin, particularly regarding animal welfare during production, transport, and slaughter. The aim of the present study was to review the factors affecting the welfare of lambs for slaughter in Chile and show the implications on meat quality. Rounding up and driving the lambs from the fields in large extensive production systems and long distance transport through difficult geographical routes affect the blood variable indicators of stress and reduces muscle glycogen reserves, increasing the risk of high pH of meat. In small farmer sheep production conditions there is a lack of appropriate installations for loading/unloading and deficiencies in vehicle structure specific for lambs; this together with the work of untrained handlers results in a high percentage of mortality and bruised carcasses, compared to European studies. These problems are common for other South American countries and should be addressed firstly by educating and training all the people involved in the lamb meat chain regarding animal welfare. In Chile there is legislation, ongoing since 2013, regarding the protection of animals during production, transport, and slaughter, including compulsory training of animal handlers and livestock transporters, which should improve animal welfare and meat quality.

## 1. Introduction

Chile has a population of 3,938,119 sheep [1], that are concentrated (64%) in the Chilean Patagonia; these sheep are mainly of the Corriedale and Merino breeds and their crosses, and are kept in large herds (thousands) in extensive grass pastures (mainly *Festuca gracillima* in the form of tussocks). The rest of the sheep are kept in the central-southern part of the country, mainly in flocks of 50 to 500 ewes owned by small farmers; they are from the Suffolk, Hampshire, and Texel breeds, are also kept on pasture (ryegrasses and legumes) but are usually collected in pens at night to protect them from predators.

Although the sheep population is small compared to China, Australia, and New Zealand, Chile has some comparative advantages in terms of the sanitary condition of livestock and potential insertion in international markets. Chilean sheep meat exports have demonstrated a great variability over-time regarding volume of product and countries of destination. Forty percent of Chilean sheep meat exports are sent to Europe (The Netherlands, Spain, and Denmark), reaching 6047 tons of frozen products in 2013 [2].

Consumers have become increasingly demanding that products of animal origin are not only healthy, natural, and palatable, but also result from animals that are produced, transported, and slaughtered in a humane way [3]. According to these consumer requirements it has become necessary to include animal welfare as part of the quality assurance schemes in Chile [4]. This requires a review of the factors that could be affecting animal welfare and their consequences, in terms of product quality.

The aim of the present study was to review the factors affecting the welfare of lambs for slaughter in Chile and show the implications on meat quality. This review is based on a series of studies, mostly concerning commercial transport, undertaken by the authors in collaboration with under- and postgraduate thesis students between 2005 and 2018. The results have been disseminated at conferences, but very few have been published in scientific journals.

## 2. Factors Affecting Animal Welfare and Meat Quality during Transport Preslaughter Operations in Lambs

Animal transportation is an important and necessary process in sheep meat production systems. It includes several stages, starting with rounding up or gathering of animals on farms, driving them to the corrals or pens close to loading ramps, loading them onto vehicles, transporting them to the slaughterhouse, unloading them, and then driving them to lairage pens and to slaughter. It is important to handle animals carefully, considering the welfare of the animals throughout the chain of events [5].

The whole process could be repeated several times if animals go through livestock markets for commercialization, instead of being sent directly from farm to slaughter. Gallo and Tadich [4] showed that selling through livestock markets or intermediate dealers is common in South American countries. Moreover, in some countries this process may be aggravated by other conditions like inadequate installations, untrained handlers, traditions, and lack of legislation.

Although in sheep the effects of poor transport and handling on the quality of the meat produced does not seem as evident as in cattle and pigs [6], geographical and human conditions in Chile, and in other countries of the region, suggest that in many occasions the welfare of lambs for slaughter is negatively affected and that this can also negatively affect meat quality and hence have economic impact [7,8].

### 2.1. Handling before Loading

Selling lambs for slaughter starts by rounding them up from the grazing fields and driving them into corrals (pens); then they are weaned and, immediately afterwards, loaded onto a vehicle. In the extensive production conditions of the Chilean Patagonia, it is common that lambs are driven with their dams for several kilometers, coming from distant fields in groups of thousands of sheep at the same time. This requires the aid of people on horseback and trained dogs (Figure 1a,b). Basically, several people on horseback and their dogs distribute themselves on the fields and start rounding up the sheep; as the sheep are bunched one driver will lead in the front showing the way towards the corrals, whilst the others remain around the back and sides of the advancing herd. The procedure can take several hours depending on the size of the farm.

Brito [9] and Tapia [10] described a duration of eight hours since the start of the collection of the animals until they arrived at the farm corrals; once they arrived they were left for some time to rest in order to wean the lambs and load them into the vehicles to be sent directly to a slaughterhouse. This process is very stressful; as can be gathered from the high concentrations of cortisol in the sheep/lambs: before loading, and as observed in different studies, the cortisol levels are two or three times higher than the resting value in lambs of the same origin and age that were catheterized in the jugular vein and accustomed to handling (Table 1). The packed cell volume and creatinphosphokinase (CK) concentrations in the sheep/lambs are also higher compared to resting values; it is noticeable that the CK concentrations in the sheep/lambs were even higher than after transport in the case of both studies in the Chilean Patagonia [11,12]. These results reveal muscle damage and dehydration before transport; moreover, lambs were often observed panting.

In the central-southern region of Chile, where farms are smaller and the distances the lambs have to be driven are shorter, although lambs are also weaned just before loading, the concentrations of the same variables are lower [13,14]. Regarding these results, a possible solution to reduce stress before loading in the Chilean Patagonia could be rounding up the animals a couple of days before the expected transport, leaving them close to the loading point.

In order to handle the lambs correctly during rounding up and driving, it is important to have trained handlers and dogs, the driving aids should be used correctly in order to avoid unnecessary stress and injury to the animals. Mera [16] described that the driving of sheep in the Chilean Patagonia was performed using devices like plastic bags, sticks, bottles with stones inside, and other rattling instruments to produce noise; trained dogs were almost always used (Figure 2). Magellan sheepdogs are the result of breeding between British sheepdogs that arrived in Chile in the late nineteenth and early twentieth centuries and have adapted to the Chilean Patagonia climate conditions [17]. They have some similarities in coat color and markings with Bearded Collies, Border Collies, and Bobtails, however, the color tones within and between the coat color patterns seen in Magellan sheepdogs show distinct features from other traditional Collie dog breeds [18].

It was often observed that some driving aids were not used correctly: hitting animals with driving aids or more commonly, grabbing lambs from different parts of the body (like ears and legs), and pulling them from the wool. This will stress and hurt the lambs and probably produce painful bruises, hence affecting animal welfare and meat quality [19]. Similar problems have been observed in the United Kingdom [20]. However, after these studies were performed, new legislation has been produced in Chile which forbids these practices since 2013 [21,22,23] and it is expected that these will slowly disappear and the handling of animals will improve.

The Chilean regulations for the protection of animals cover livestock in industrial production systems [22], livestock transport [21], and slaughter [23]; the general aim in all cases is to acknowledge that farm animals are sentient beings and that there is a need to reduce pain and unnecessary suffering in them. The regulation for industrial production systems [22] includes indications regarding feeding, health, and housing conditions; there are specific prohibitions during handling like hitting, pricking, or grabbing animals from different parts of the body and there are training obligations. These make it compulsory to have at least one person trained as an animal handler in each productive system, which could be the farmer themself or a designated person with a training certificate. The training consists of a 16-h course given by educational institutions that are recognized for this purpose by the competent authority (the Agriculture and Livestock Service (SAG) of the Chilean Ministry of Agriculture). Enforcement is structured in the way of audits by SAG, but is still at an early stage at farm level and more effort has been made by enforcement regarding livestock transporters and slaughterhouses. However, efforts are being made by producer associations, educational institutions, and the SAG to develop and spread information on good handling practices in order to improve animal welfare and create awareness on the legislation.

### 2.2. Loading and Unloading

Loading of animals onto a transport vehicle is one of the most stressful procedures and a critical point within the general transport process [24,25]. In order to facilitate loading, it is important that animals are calm, that the stockmen are trained in animal handling, and that the infrastructure is adequate, particularly loading ramps. Considering that lambs are transported mainly in two or three floor vehicles, a frequently observed problem is that ramps can be very steep when loading the compartments of the upper floor [26]. Mera [16] and Tarumán [27,28] found that in the farms of the Chilean Patagonia loading of animals was generally done calmly using the appropriate ramps and facilities (Figure 3a,b). However in the central-southern part of the country, it was commonly observed that small farmers did not have proper loading facilities for sheep at all, and lambs were loaded one by one manually, often pulled by the wool and other parts of the body (Figure 4a,b), or even thrown and dragged [13].

Regarding the unloading of lambs at slaughterhouses, the same aspects for loading should be considered. Special care should be taken when unloading vehicles of two or three floors. Most slaughterhouses, where unloading of lambs was observed, had mobile metallic ramps (Figure 5) with rubber flooring to reduce the slipping of animals [16,26], but others only had metallic surfaces, which can easily become slippery. In some local non-export slaughterhouses there was no ramp at all for second floors and lambs had to jump or were manually unloaded one by one (Figure 4c); this increases the risk of lesions and adds more work and difficulty to the process of unloading.

In order to improve animal welfare and lamb meat quality in Chile, as well as in South America in general, it is necessary to improve farm facilities at the level of small producers, and train stockmen and transporters about behavioral principles and proper handling of lambs to reduce stress, as well as about the consequences of bad handling on product quality. Further enforcement of the legislation for the protection of animals during transport [21] should lead to improvements, as it is now compulsory that all livestock drivers must attend an official training course that considers the above mentioned aspects.

### 2.3. Transport Vehicles

Providing appropriate vehicles for livestock transport that are built and equipped according to the specifications of the species, size, age, and physiological conditions of the animals transported is an unquestionable principle for the protection of animals during transport [29]. The European Commission [30] established that vehicles must warrant security to the animals, avoiding the escape or fall of animals, and protect animals from adverse environmental conditions; they should be easily cleaned and disinfected, have a proper floor to reduce the leakage of urine and feces, have adaptable compartments according to the type, size, and needs of the animals and be resistant enough for the weight carried.

Chilean Patagonia sheep transport vehicles (Figure 6a,b) are commonly three floor trucks with a trailer, have internal divisions that afford between eight and 12 compartments (pens) per floor; floors are mostly metallic and some wooden, the walls are smooth without elements that could produce injuries to the animals, and the metallic floors have a collection system that avoids urine and feces leaking to the lower floors [16,26,28,31]. Ventilation is provided by the absence of roofing and the presence of lateral open spaces on the walls in each floor. The fact that they do not have a roof or any other protection can be a problem when there are adverse climatic conditions like rain and snow [28]. The described livestock vehicles are exclusively used for the transport of sheep and, therefore, are only used during the lamb production season, which can last approximately seven months [27,28].

The livestock vehicles used for lambs in the central-southern part of Chile are much more rudimentary. Two floor vehicles with a trailer are more common in this region, usually used primarily for the transport of pigs [32]; these have between zero and six compartments per floor. In the case of the small farmers of this region, they use the same vehicles for cattle [13]: no vehicles are built specially for the transport of sheep. Therefore, the vehicles in this region have many structural deficiencies such as bad ventilation and the, sometimes improvised, wooden second floors do not avoid the leakage of urine and feces from upper floors. These conditions are similar to those existing in many other South American countries, except for the large producers in Argentina and Uruguay, and do not provide adequate conditions for the transport of lambs [4]. 

### 2.4. Space Allowance during Transport

Stocking density during transport can be defined as a certain amount of live weight of animals within a vehicle compartment; it is expressed as kg live weight per square meter [33]. Nowadays it is preferred to use the term ‘space allowance’ instead of ‘stocking density’ [34], because it better reflects conditions from the point of view of the animal: it is the space an animal can use during transport. It has been recommended that space allowance should be enough for all lambs to lie down [35], particularly for journeys longer than 4 h, which is the most common journey length in the Chilean Patagonia. Low space allowances lead to fatigue in those lambs that are forced to stand instead of lie down during transportation, and that can cause muscle damage and an increase in the plasma concentrations of CK [36,37,38]. With little space, animals are hindered in making the necessary position changes to adjust their posture while the vehicle is moving.

The space allowance recommended by the European Commission [30] for sheep during transport longer than 4 h is 0.8 m^2^/100 kg lamb. In the Chilean Patagonia, where the live weight of lambs is between 28 and 32 kg, Tarumán and Gallo [31] and Strappini et al. [26] observed space allowances of 0.66 m^2^/100 kg per unshorn lamb, for journeys between 4 and 12 h. Mera [16] and Carter and Gallo [39] found allowances of 0.75 m^2^/100 kg per unshorn lamb for journeys between 46 and 75 h (terrestrial–maritime transport). Figure 7a,b shows the space allowance at which lambs arrive at the slaughterhouses in the Chilean Patagonia. In the central-southern region of Chile, the live weight of lambs is usually between 40 and 45 kg; Castro [32] registered a mean space allowance of 0.7 m^2^/100 kg lamb, however a high variability between loads was observed due to the fact that transport vehicles do not take a full load of lambs to the slaughterhouse, because lambs are produced mainly by smallholders [13], and they do not use separations or transport pens (Figure 7c).

A decrease in terms of space allowance was observed over the season [28], a fact that is consistent with lambs becoming older and heavier. This shows that lamb transporters mainly take into consideration a certain number of lambs per pen, disregarding the size of different breeds, actual live weights, and even the duration of the journey. The fact that space allowances are below or at the limit of those recommended by the European Commission [30] is due to economic reasons, because transporters and producers try to transport the maximum possible number of lambs per journey to reduce costs, but do not take into consideration animal welfare nor the final quantity and quality of the product.

In order to try to make improvements to the long distance commercial transport of lambs, Navarro [11] transported, experimentally, 30 kg live weight lambs at the low space allowance used commercially (0.66 m^2^/100 kg lamb) and at a higher space allowance (1.0 m^2^/100 kg lamb), using two pens in the same vehicle; the experiment was repeated four times during the long distance terrestrial–maritime commercial transport. Lambs at the higher space allowance were also provided with drinking water, in specially designed troughs. The groups of lambs with more space registered a higher frequency of individuals that lay down during the journey, but also more lambs walked and slipped (Table 2). Interestingly, lambs at the higher space allowance did drink water during the journey and finally produced higher carcass weights than their controls; however, they had more bruised carcasses (70% vs. 60% [40]).

### 2.5. Distance of Travel and Journey Duration

The Chilean Patagonia produces most of the lambs in Chile; in the regions of Magallanes and the island of Tierra del Fuego (Figure 8) most of the lambs produced are slaughtered in local export slaughterhouses, and the distance of transport fluctuates between 15 and 300 km only [27,28]. However, journey duration for these short distances was found to fluctuate between 3 and 12 h. The duration of the journeys is not directly related to the distances travelled, due to the existence of unpaved roads in bad condition, many also existing in difficult geographical conditions, including short ferry crossings between the island of Tierra del Fuego and the continent [26,28]. In Chile the farms are connected through gravel roads with secondary paved roads, as is the case in many other South American countries [4]. Unpaved roads are often in poor condition, and besides increasing the transport time, they produce unwanted extra movement of the vehicle, which increases fear in the lambs and reduces their chance of maintaining balance. Ruiz de la Torre et al. [41], registered the effects of a moving vehicle on good and bad state roads, finding that the concentration of cortisol, heart rate, and pH was more favorable on good roads. Cockram et al. [42] compared sheep transported on motorways versus secondary roads, finding that the latter showed more balance losses, less number of animals lying down ruminating, and a greater level of disturbance among animals.

The region of Aysén, also in the Chilean Patagonia (Figure 8), is one of the most isolated in Chile and does not have continuous terrestrial connectivity with the rest of the country. Moreover, it does not have an export slaughterhouse and there is insufficient infrastructure to slaughter all the lambs produced locally. Therefore, for many large producers there is no other option than to transport their lambs to slaughterhouses located in the central-southern part of the country. This means that each year approximately 20,000 lambs are transported for journeys lasting 48 to 75 h (830–1350 km), including a long maritime section (22–36 h) on roll-on roll-off ferries (Figure 8), and no possibility for resting stops [19]. Comparatively, when lambs of this same region are slaughtered locally, they are transported up to 400 km, in journeys lasting 0.5 to 10 h; there is no correlation between distance travelled and journey duration either [16,43].

According to a study of 47,365 lambs arriving during one season at the two main export slaughterhouses of the central-southern region of Chile [32], the transport duration for most loads varied between 1 and 12 h for distances of up to 800 km (excluding some loads arriving from long distance terrestrial–maritime transport). In the case of local small producers, who take their lambs to local non-export slaughterhouses, the journeys last less than 4 h [13].

In the case of transport longer than 24 h, it was never observed that lambs received water or feed during these journeys [16]. Table 1 shows that the packed cell volume of lambs transported for over 24 h [12] increased compared to basal values, and also to short transport [13], which reflects dehydration. According to Warriss [44], lambs do not drink water while the vehicle is in motion, but they do during rest stops. Navarro [11] found that during sea crossing, while the vehicle was on the ferry, the lambs did drink water (Table 2), showing that it would be good from an animal welfare point of view to provide water in these journeys. The increased concentration of betahydroxybutyrate in the lambs after long distance transport (Table 1) suggests that lambs should also be fed during these journeys [12].

The results presented here in terms of distances travelled and journey times for lambs transported to slaughter in Chile differ completely from those in Europe [6,45], mainly due to geographical conditions and road types. Distances travelled are much longer, but moreover, for the same distance the vehicles take longer to reach destiny due to difficult road conditions. This situation is similar in other South American countries [4]. At present, with the new legislation for the protection of animals during transport in Chile [21], it is compulsory to provide animals with water and food for journeys longer than 24 h, and this should partly counteract the negative effects of these journeys on animal welfare, and reduce losses in terms of live and carcass weight, and meat quality of the lambs shown in Chilean studies [11,12,39,40].

### 2.6. Driving Skills

The level of driving skill can also have an influence on the welfare of the transported animals, affecting the risk of lesions and disturbing the ability of sheep to rest. Training the transporters to drive carefully is crucial for handling animals without producing fear [46]. Up until 2013 very few livestock transporters had had formal training in animal handling and animal welfare in Chile [16,26,28]. In 2011, Castro [32] found that in the central-southern part of Chile, some lamb transporters had had at least informal training in animal handling. Since 2013, when the legislation on livestock transport was implemented [21], training has become compulsory and transporters have to carry a certificate of formal 16 h training, given by an institution recognized by the competent authority (The Agriculture and Livestock Service of Chile).

In general, in South America the amount of legislation on livestock transport has increased in many countries [47] and there is increasing awareness about animal transport. This should slowly improve the driving skills of livestock transporters and the welfare of animals, although much more of the emphasis on enforcement has been put on cattle than on sheep. On the other hand, Grandin [48] showed that there was an increase in the welfare of animals when livestock drivers received a bonus payment for meat quality of the animals transported. This strategy could also help improve welfare and meat quality.

### 2.7. Lairage at the Slaughterhouse

Once the lambs are unloaded at a slaughterhouse, they are driven to the lairage pens to rest. During lairage they are provided with water, but no feed, unless they have to remain more than 24 h before being slaughtered; in this case they have to be fed according to Chilean regulations [49]. The duration and conditions of lairage are important factors that can add more stress to the transport and preslaughter operations. Lairage should allow the animals to recover from transport stress and rehydrate [50]. However, the World Organization for Animal Health [34] recommends that animals should be slaughtered as soon as possible after arrival.

In Chile, lambs usually arrive at the slaughterhouse the evening before being slaughtered, and they remain in lairage for over 16 h [16,28,43]. According to the same authors, considering that most lambs are transported for between 4 and 12 h, this long lairage increases the time animals are deprived of feed, because the obligation of feeding after remaining 24 h at the slaughterhouse does not include the transport time before arrival.

In the case of long distance transport (>24 h), total fasting time for lambs can reach up to 74.5 h, since the moment of collection of the animals on the farms and the actual slaughter [9,12]. A long lairage is detrimental to animal welfare because it increases the risks of suffering from stress due to hunger, thirst, and other adverse climatic conditions such as lack of sufficient and comfortable space to lie down, fights, and accidents. On the other hand, all of the aforementioned factors will make lambs more susceptible to higher weight losses, sustaining bruises, and higher muscle pH [7,40]. Efforts should be made to reduce the time lambs have to remain in lairage in order to improve animal welfare and meat quality; this mainly requires better planification of the transport and slaughter operations.

## 3. Consequences of Transport and Preslaughter Operations on Animal Welfare and Meat Quality in Lambs

The effects of transport and related preslaughter operations on the welfare of animals can be measured directly on the individuals through changes in both their behavior and physiological indicators [37,38,51]. However, often the welfare conditions of animals destined for slaughter are also evaluated by determining mortality, weight loss, frequency of sick or injured animals at arrival, and physical and biochemical defects in their carcasses, such as the presence of bruises and alterations of muscle pH, color, and water retention [19,52].

In Chile, as in South America in general, studies in cattle are mainly concentrated on looking at the quantitative and qualitative effects of transport and handling on the meat products, because these measures are easy to use and have direct economic consequences on the producers; therefore, they are easier to demonstrate to the handlers and producers [7,8,53,54]. The study of the consequences of transport and preslaughter operations on the quality of lamb meat could also provide information that might be useful to make improvements both in animal welfare and meat quality.

### 3.1. Mortality

Death of animals is the worst consequence that inadequate handling and transport previous to slaughter can have and provides clear evidence of poor welfare during the process. Death will occur particularly in those animals with previously compromised health due to the additional stress of the handling [6,55]. Knowles et al. [55] reported a mortality of 0.007% in lambs transported directly from farm to slaughter (62 miles) in the south of England and 0.031% in those going through the market (199 miles). In Chile mortalities of 0.1 to 0.13% have been reported at arrival of commercial loads of lambs at the slaughterhouses [16,31,32]. These higher mortalities are probably related to some of the factors discussed previously, such as the much longer distances (and time) travelled by the lambs, the low space allowances, bad roads, use of inadequate vehicles, and untrained handlers. Mortalities could be reduced by improving slaughterhouse infrastructure regionally, so that lambs do not need to be transported over large distances, or by improving the transport conditions, providing water and feed, as well as higher space allowance during the long journeys.

### 3.2. Weight Loss

From the moment sheep are collected from the fields to be loaded and transported they are, in general, deprived of water and feed; when arriving at the slaughterhouse they are provided with water but continue without feed during lairage. Deprivation of feed and water in animals produce changes in the blood variables related to stress and inanition [38], and also changes in behavior, due to hunger and thirst. All the aforementioned indicators reflect that animal welfare is compromised, being of ethical concern. In quantitative and economical terms, lambs will reduce their live weight, and if transport and food deprivation is long enough, they may also reduce carcass weight [39].

To investigate the weight losses in lambs, four commercial loads transporting 2106 lambs (mean 29 kg live weight), bred and loaded at the same farm were followed and studied; two of these loads of lambs were slaughtered at a local slaughterhouse after a terrestrial journey of 12 h and the other two loads were transported to a slaughterhouse in the central-southern region of the country, after a 46 h of terrestrial plus maritime ferry crossing journey [39]. In each load, 25 randomly chosen lambs were individualized to measure live weight before and after transport, hot carcass weight, bruising, pH, and muscle glycogen (total 100 lambs). In the lambs transported for 46 h the live weight losses were higher (13.4 vs. 4.8%) and the carcass weights were lower (13.7 vs. 14.9 kg) than in the lambs transported for 12 h. Considering that around 500 lambs are transported in a load (truck with trailer, three floors), and each lamb loses approximately 1 kg of carcass weight, approximately 500 kg of lamb carcasses are being wasted in one of these journeys. Certainly, this cannot be sustainable in economic terms, nor in animal welfare terms.

### 3.3. Bruising

In sheep, as in cattle, bruises are not visible in the live animal due to the thickness of the skin and wool, hence they can only be detected at postmortem inspection of the carcass [56]. The presence or absence of bruises on a carcass, as well their severity and extensivity, are indicators of the welfare of the animal preslaughter, because bruises are associated with the process of transport and handling of the animals [56,57,58,59]. Besides being painful while the animal is alive, the damage produced by bruises on the carcass requires trimming and parts of the carcasses have to be condemned, reducing the weight and value of the product [19,60].

Knowles et al. [55] found that in lambs, the distance travelled was a poor predictor for bruising, but other studies have found a positive relationship between these variables [6,61,62].

The percentage of bruised carcasses found in Chilean studies with long transport distance varies between 33% [39] and 64% [40], and for lambs transported up to 12 h it varies between 22.7% [63] and 25% [40]. Tarumán et al. [63] found a significant (*p* < 0.05) association of transport time and live weight with bruising, with the most exposed animals being those transported for more than 4 h and those that were heavier. The latter could be due to the higher actual stocking density found for lambs of higher weights during transport, because transporters consider number of lambs per pen instead of kg live weight per pen [28].

The characteristics of the bruises in lamb carcasses were classified by Tarumán and Gallo [31] according to severity (or depth in terms of tissues affected: subcutaneous 1, muscular 2, and bone or fracture 3), to extension of the lesion (diameter), and the anatomical region affected.

Due to the small size of lambs, there is a risk of lesions in the transport vehicles as their legs could be easily trapped in floor or wall cracks in the vehicle and also open spaces between the vehicle and the ramps. According to Anderson and Horder [64], external factors like transport and handling would be responsible for the location of a bruise. For instance, during loading and unloading, lambs are often pulled by the wool or grabbed by different parts of the body when they refuse to move, which also increases the likelihood of bruising [16,28]. This handling is in accordance with the small size of most bruises observed (<5 cm) and the tissues affected (mainly subcutaneous). These findings are similar to other international studies that have mostly registered superficial, small bruises: 2–4 cm in lamb carcasses [62,65].

Several authors in Chile [31,40,63] have found that the loin is the predominant anatomical location of the bruises in lambs, which agrees with Cockram and Lee [61]. The location of the bruises on the loin could be explained by handling, due to handlers pulling the lambs from the wool on their back. In fact Jarvis and Cockram [20] studied the relationship between potentially traumatic events during loading, unloading, and handling of sheep, and found a relationship between the occurrence of wool pulls by handlers and bruising on the sheep.

Tarumán et al. [63] found that there is a risk for more bruises when lambs are transported commercially on bad roads, as shown earlier by Ruiz de la Torre et al. [41] experimentally. Large lesions (diameter) and deep lesions (affecting muscle) are more likely to be produced when lambs fall during transport and are then trampled by other animals standing.

Considering that compared to international studies there is a high prevalence of bruised lamb carcasses but the lesions are mainly small and superficial, it is suggested that handling the animals appropriately by training stockmen and transporters could reduce this problem.

### 3.4. pH Alterations and Other Defects

Meat pH is one of the main measures used to monitor quality, however, in Chile it is not a routine measurement at lamb slaughterhouses. Physical and physiological stress of animals before slaughter in general affects animal welfare and also meat pH [66]. Factors like high stocking densities during transport to slaughter [67], duration and conditions of lairage [68], adverse climatic conditions, social disruption, or a novel environment can cause pH decrease [69]. Therefore a high final pH (usually measured at 24 h postmortem) in lamb carcasses, will be a reflection of the inadequate conditions and the stress that animals have undergone previous to slaughter and can be used as a postmortem welfare indicator.

Watanabe et al. [70] categorize lamb carcass pH in three groups: normal or low (<5.8), intermediate (5.8–6.3), and high (>6.3) and state that meat with a pH between 6.2 and 7.0 will be dark, firm, and dry, useful only for manufacture. According to this, Pantanalli [43] categorized 27,697 lamb carcasses at one slaughterhouse in the Chilean Patagonia, measuring pH in the leg (between the Semimembranosus and Semitendinosus muscles). This author found that 67% of the carcasses had a normal pH, 32.9% fell in the intermediate category, and only 0.1% had high pH. In another slaughterhouse, Tarumán et al. [63] also measured pH in the leg of 1150 lamb carcasses and found that 45% fell in the intermediate range and none in the range of high pH. The latter authors found that transport time and presence and number of bruises were factors that had a significant association with pH (*p* < 0.05). Due to this association, both bruising and meat pH could be used at slaughter plants as indicators of meat quality, and also of animal welfare.

Table 3 shows the concentrations of muscle glycogen and the mean pH values found in several studies in Chile, where both variables were measured in the Longissimus thoracis muscle of lamb carcasses. The mean values of pH were similar in all studies and below 5.8, hence could be classified within the normal range according to Watanabe et al. [70]. In the study of Vargas [40], who transported lambs in four journeys that lasted between 32 and 42 h, 48% of the carcasses were categorized as intermediate pH according to Watanabe et al. [70] and none in the high pH category. In the case of the study of Carter and Gallo [39], 50% of the carcasses of lambs transported for 12 h and 40% of those transported for 46 h had intermediate pH and none had high pH. However, in general, there was no clear relationship between transport time, pH, and concentration of muscle glycogen. It is possible that the lack of relationship is due to the fact that there was a great variability between studies in the total fasting time. In the case of Carter and Gallo [39] there was no difference in terms of pH nor muscle glycogen between the lambs subjected to 12 or 46 h transport; these lambs had been deprived of food for 8 h before transport (on farm) and then for 6 h after transport, during lairage; in farm the lambs had been driven from the fields, hence exercising, and then weaned. The low concentrations of muscle glycogen found in both groups are in fact similar to the 11.1 µmol/g found by Bond et al. [71] in lambs exercised for 24 h. Jacob et al. [68] found that the effect of stress during transport may be different than stress due to exercise, and travel time was correlated positively with glycogen concentration in the Semimembranosus muscle whereas farm curfew time was correlated negatively with glycogen concentration in the Semitendinosus muscle. So it could be that the exercise, due to the long walk during mustering and driving, and the stress of weaning had already depleted the glycogen reserves of sucker lambs before transport in the case of Carter and Gallo [39]. In the study of Vargas [40], the lambs used had been weaned more than a week before the experiment and were collected from fields close to the loading point, and then transported for 37 h and slaughtered immediately after arrival (short lairage) or after 20 h lairage (long lairage). In the study of Baeza [13], the 10 loads of lambs were transported for 0.5 to 4 h only, with a total fasting time between 16 and 28 h, considering the time before loading and lairage at the slaughterhouse. The concentrations of muscle glycogen found by Vargas [40] and Baeza [13], shown in Table 3, are closer to those published by Lowe et al. [72] in lambs on pasture (25.4 and 42.5 µmol/g) and by Jacob et al. [68]. The latter also concluded that lairage time did not have a clear effect on pH and glycogen concentrations, mainly because the consignments of lambs studied also had different origins, transport times, and different times off feed on the farms before loading. However, sucker lambs in general had lower glycogen concentrations in muscle and higher pH than carry-over lambs (weaned and older).

In order to try and elucidate the effect of fasting on muscle glycogen, we [73] measured the concentration of muscle and liver glycogen experimentally in five muscles of eighteen weaned lambs of similar characteristics to those used in the Chilean Patagonia commercial studies. The lambs were kept in a pen and slaughtered after increasing time without feed (with access to water). The results are shown in Table 4. Glycogen content was higher and depletion due to fasting was faster in liver than in muscle. Muscle glycogen concentrations decreased in general in all muscles with increasing fasting time between 0 and 56 h, and the *Longissimus thoracis* (LT) muscle showed the highest values. The mean concentration of muscle glycogen in LT at 32 h of fasting was similar to the values reported by Vargas [40] and Baeza [13] in lambs with similar total fasting times, but still much higher than the values reported by Carter and Gallo [39].

In general, the relationship between muscle glycogen concentration and pH in lambs seems less clear that in cattle, where many more studies have been performed and pH is regularly measured at slaughterhouses, which is not the case in Chile at least. Notwithstanding this, within the practical recommendations to address high pH problems are reduction of exercise and stress to the minimum possible on farm before transport, and reducing transport and lairage times, because this stops lambs from using their energy reserves before being slaughtered. In Chile, as in many South American countries, lamb is still mainly consumed fresh (refrigerated) and for special occasions only; therefore, there has not been much research in terms of the technological quality of lamb meat for further processing or the use of techniques like vacuum packaging to increase shelf life. The development of new techniques will require further knowledge about the biochemical processes involved in the transformation of muscle to meat in lambs and should also help improving lamb welfare and meat quality.

## 4. Conclusions

Rounding up and driving suckling lambs from distant fields in extensive sheep farming systems, like the Chilean Patagonia, is stressful and reduces energy reserves of the animals; a possible solution to reduce stress before loading in these cases could be rounding up the animals some days before the expected transport, leaving them close to the loading point.

Although facilities for the loading and transport of lambs are appropriate in the Chilean Patagonia, deficiencies were observed at the level of small producers particularly in the central-southern region; creating awareness to farmers and transporters about animal welfare and the appropriate infrastructure; developing good handling practices at this level should receive assistance by governmental institutions.

Mishandling the lambs, as it was often observed when using driving aids incorrectly, grabbing them from different parts of the body or pulling them from the wool, not only has consequences on the welfare of animals but also on product quality in Chile, as determined by the high percentage of bruised carcasses. Further enforcement of the legislation for the protection of animals in industrial sheep farming systems and during transport should lead to improvements in this respect.

The duration of lamb transport in Chile is often increased due to geographical reasons and bad roads; the space allowances used for lambs are similar in long and short distance transport. The dehydration and increased concentration of betahydroxybutyrate in the lambs after transport of 24 h confirm that lambs should be fed and watered during these journeys. More emphasis on enforcement of the law is needed to improve the welfare and reduce losses in terms of mortalities, live and carcass weight, as well as meat quality of lambs.

Efforts should be made to reduce the time lambs have to remain in lairage at the slaughterhouses in order to improve animal welfare and meat quality; although the relationship between muscle glycogen concentration and pH in lambs seems less clear than in cattle. Within the practical recommendations to address high pH problems in Chile are reduction of exercise and stress to the minimum possible on farm before transport and reducing transport and lairage times. This stops lambs from using their energy reserves before being slaughtered and mainly requires efforts in better planning of the timing of transport and slaughter operations. Meat pH measurements and bruising should be used at slaughter plants as indicators of meat quality and also of animal welfare.

## Figures and Tables

**Figure 1 animals-08-00165-f001:**
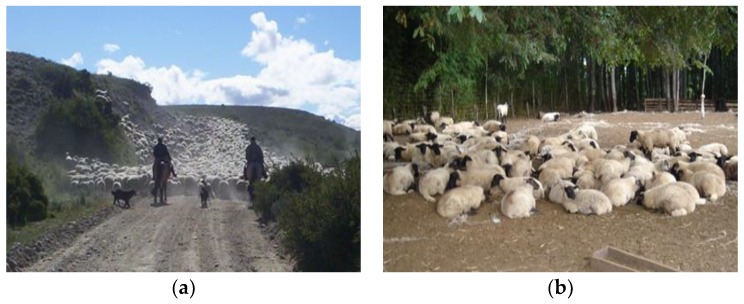
(**a**) Rounding up and driving sheep in the Chilean Patagonia; (**b**) lambs resting in a pen before loading in small farmer conditions of central-southern Chile.

**Figure 2 animals-08-00165-f002:**
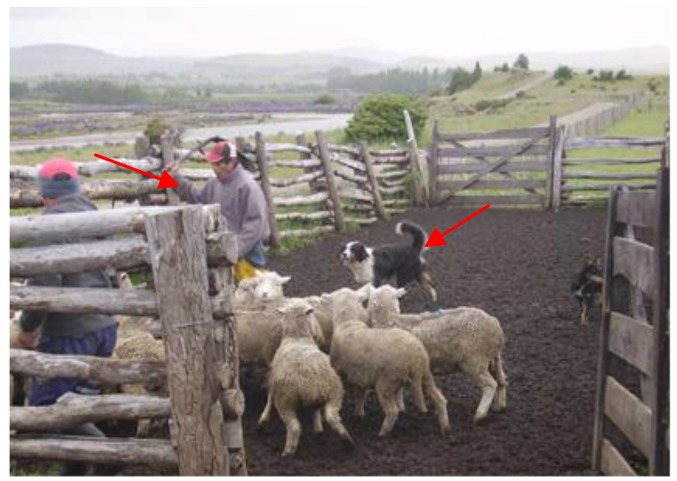
Moving sheep with dogs and other driving aids.

**Figure 3 animals-08-00165-f003:**
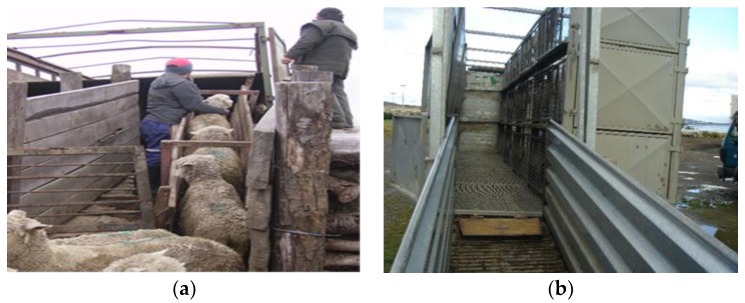
(**a**) Loading sheep using a wooden ramp; (**b**) truck with metallic adjustable pens and double flooring, also showing the metallic loading ramp.

**Figure 4 animals-08-00165-f004:**
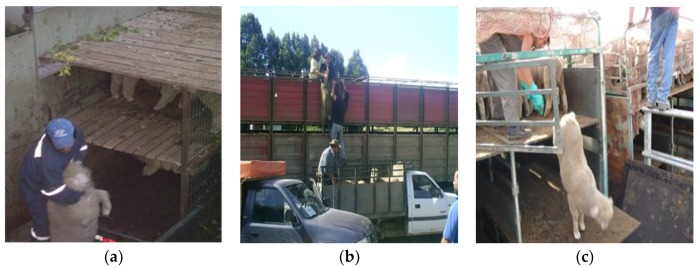
Inadequate practices for loading (**a**,**b**) and unloading lambs (**c**).

**Figure 5 animals-08-00165-f005:**
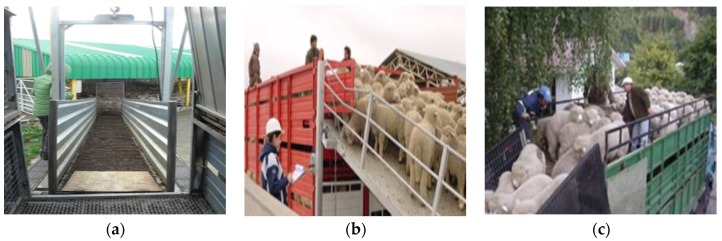
(**a**–**c**) Unloading lambs at the slaughterhouse using mobile metallic ramps.

**Figure 6 animals-08-00165-f006:**
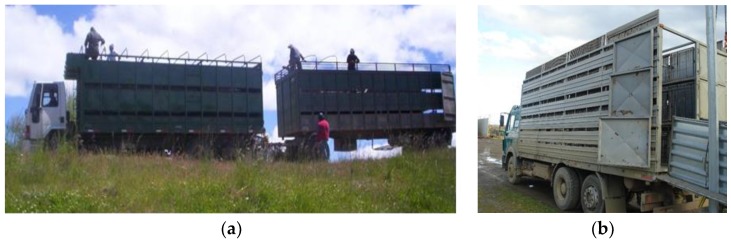
(**a**,**b**) Typical vehicles used in the Chilean Patagonia for the transport of lambs.

**Figure 7 animals-08-00165-f007:**
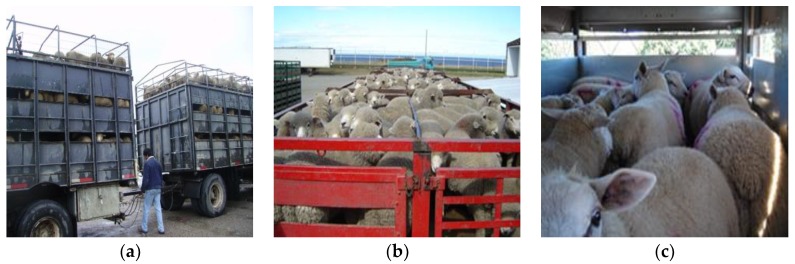
(**a**,**b**) Vehicles loaded with lambs at space allowances of approximately 0.66 m^2^/100 kg; (**c**) small trailer carrying lambs produced by small farmers at approximately 0.8 m^2^/100 kg.

**Figure 8 animals-08-00165-f008:**
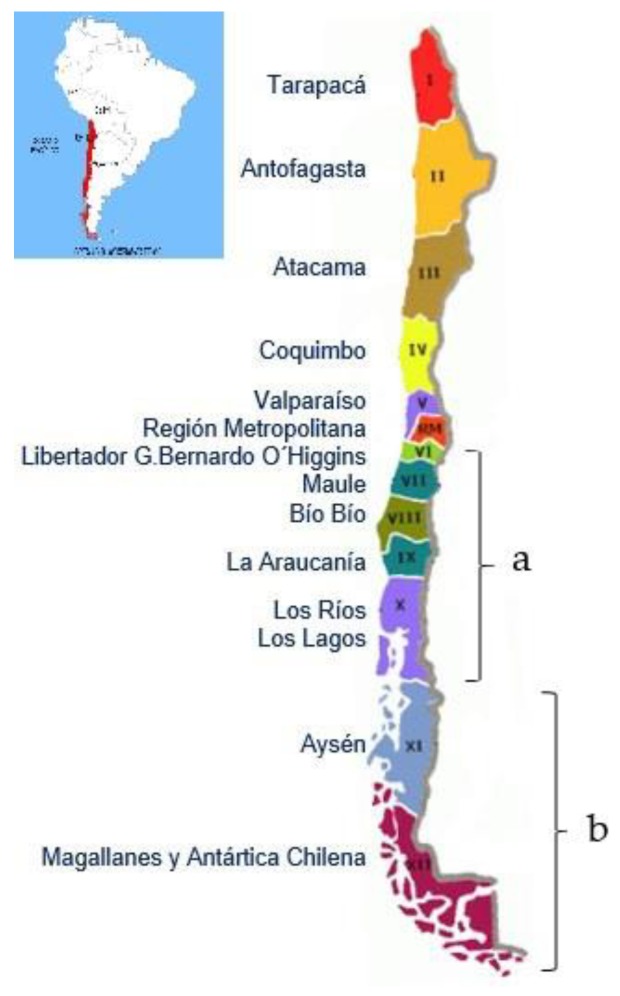
Map of Chile showing the central-southern region (**a**) and the Chilean Patagonia (**b**).

**Table 1 animals-08-00165-t001:** Blood variable indicators of stress in lambs subjected to different conditions on farm and different transport durations.

	Barrientos [15]	Navarro [11] Transport = 49 h				Tadich et al. [12] Transport = 48 h		Baeza [13] Transport = 0.5–4 h		Larrondo et al. [14] Transport = 3 h	
	Resting Values	On Farm		After Transport		On Farm	After Transport	On Farm	After Transport	On Farm	After Transport
		Low ^1^	High	Low ^1^	High						
**Cortisol (μg/dL)**	0.9 ± 0.4	2.2 ± 0.17	2.26 ± 0.25	3.72 ± 0.29	3.39 ± 0.27	2.7 ± 0.2	3.9 ± 0.2	1.68 ± 1.56	1.57 ± 1.32	4.72 ± 1.47	3.3 ± 1.75
**Glucose (mmol/L)**	3.9 ± 1.36	-		-		4.5 ± 0.11	4.1 ± 0.08	-	-	5.28 ± 0.77	5.02 ± 1.23
**Lactate (mmol/L)**	0.7 ± 0.31	-		-		4.8 ± 0.2	3.3 ± 0.2	-	-	-	-
**CK (U/L)**	180 ± 76.5	995 ± 186.64	992 ± 197.42	448 ± 63.37	432 ± 47.05	705.5 ± 49.2	635.9 ± 40.1	166.9 ± 117.2	144.5 ± 123.8	325.5 ± 143	422.9 ± 234
**PCV (%)**	36.6 ± 2.2	42 ± 0.58	45 ± 0.28	44 ± 0	44 ± 0.28	43.3 ± 0.4	43.1 ± 0.3	36.9 ± 4.4	34.7 ± 4.4	-	-
**B-OHB (mmol/L)**	0.3 ± 0.11	-		-		0.4 ± 0.02	0.7 ± 0.02	0.33 ± 0.12	0.5 ± 0.3	-	-

* Mean and standard errors; ^1^ Low = space allowance of 0.66 m^2^/100 kg lamb, without water supply; High = space allowance of 1.0 m^2^/100 kg lamb and with access to drinking water during the journey, lambs 30 kg live weight; Barrientos [15]: resting lambs, accustomed to handling and catheterized in the jugular vein; Navarro [11]: lambs weaned a month before transport; Tadich et al. [12]: lambs underwent an 8 h walk before being weaned and transported, Chilean Patagonia conditions; Baeza [13]: 10 groups of lambs left overnight with dams in pens, weaned, and then transported, central-southern small farmer conditions (50 to 100 sheep); Larrondo et al. [14]: lambs collected from pasture and weaned just before transport, central-southern large farmer (>500 sheep).

**Table 2 animals-08-00165-t002:** Frequencies of behaviors observed in lambs of 30 kg live weight subjected to prolonged terrestrial–maritime transport in four journeys (30–49 h) at two space allowances: Low = 0.66 m^2^/100 kg lamb, without water supply; High = 1.0 m^2^/100 kg lamb and with access to drinking water during the journey. Adapted from Navarro [11].

	Lying Down (%)		Standing (%)		Walking (%)		Slipping (%)		Drinking (%)
	Low	High	Low	High	Low	High	Low	High	High
1	7 ^a^	23 ^b^	90 ^a^	69 ^b^	1 ^a^	5 ^b^	0 ^a^	1 ^b^	1
2	40 ^a^	53 ^b^	60 ^a^	42 ^b^	0 ^a^	4 ^b^	0 ^a^	0 ^a^	2
3	22 ^a^	39 ^b^	77 ^a^	53 ^b^	0 ^a^	5 ^b^	0 ^a^	1 ^b^	2
4	12 ^a^	26 ^b^	78 ^a^	63 ^b^	0 ^a^	3 ^b^	0 ^a^	0 ^a^	2

Different letters within a row indicate significant differences (*p* < 0.05) between low and high space allowances.

**Table 3 animals-08-00165-t003:** Muscle and liver glycogen concentrations, and pH at 24 h postmortem in Longissimus thoracis muscle of lambs in different Chilean studies.

Variables	Carter & Gallo [39] Mean ± SE		Vargas [40] Mean ± SE				Baeza [13] Mean ± SE
	12 h Transport	46 h Transport	49 h Transport				0.5–4 h Transport
	0.22 m^2^/Animal	0.22 m^2^/Animal	0.2 m^2^/Animal		0.33 m^2^/Animal		-
	2–4 h Lairage	6–12 h Lairage	1 h Lairage	16–24 h Lairage	1 h Lairage	16–24 h Lairage	16–20 h Lairage
Muscle glycogen (μmol/g)	6.8 ± 5.5	5.1 ± 4.4	25.45 ± 10.06	26.84 ± 9.19	27.52 ± 12.22	30.07 ± 9.7	33.8 ± 9.1
Liver glycogen (μmol/g)	6.9 ± 12.1	5.2 ± 9.0	1.70 ± 4.43	0.00 ± 0.00	1.95 ± 3.31	0.16 ± 1.36	-
pH 24 h *Longissimus thoracis*	5.76 ± 0.20	5.75 ± 0.17	5.77 ± 0.05	5.73 ± 0.09	5.75 ± 0.07	5.70 ± 0.09	5.79 ± 0.2

**Table 4 animals-08-00165-t004:** Mean (±SD) concentrations of glycogen (µmol/g) in liver and six muscles of lambs slaughtered at different fasting times (h). Adapted from Carter and Gallo [73].

Fasting Time (h)	Hepatic Glycogen	*Longissimus thoracis*	*Semispinalis capitis*	*Supraspinatus*	*Semimembranosus*	*Semitendinosus*
0	131.12 ± 56.66	40.22 ± 8.31	18.06 ± 8.11	37.57 ± 8.77	28.78 ± 11.82	21.75 ± 3.80
6	156.09 ± 56.66	36.55 ± 8.31	19.93 ± 6.77	30.52 ± 8.77	29.56 ± 11.82	17.20 ± 3.80
20	118.04 ± 34.03	36.07 ± 9.80	22.11 ± 2.34	33.48 ± 6.09	30.54 ± 6.73	19.89 ± 13.87
32	68.24 ± 96.60	26.82 ± 18.66	20.99 ± 7.49	26.73 ± 15.99	22.59 ± 19.95	22.88 ± 12.07
44	7.87 ± 10.75	21.38 ± 4.15	15.17 ± 1.78	27.45 ± 11.53	25.89 ± 10.65	21.08 ± 14.17
56	47.03 ± 31.27	26.10 ± 11.53	15.81 ± 8.32	30.47 ± 7.91	26.49 ± 4.08	15.24 ± 11.38

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
