# Peer review of "Main Factors Affecting Animal Welfare and Meat Quality in Lambs for Slaughter in Chile"

_animals, 2018, doi:10.3390/ani8100165_

Round 1
Reviewer 1 Report
Moderate English changes required
I checked this box with some hesitation. Overall I appreciate the authors bringing the essential findings of previous studies not published in English to the English speaking community.
Although I might phrase something a different way having English as my first language does not entitle me to recommend that change to others. Saying something in a "different" way is not a value added exercise. I understood this paper completely as written (there is a couple of spelling errors) and out of respect for the authors I would not as a rule nit-pick English language use. In my opinion as a reader to put English in another's mouth is an offensive act of whiteness or being a Gringo if that is proper use of the word. This paper is the creative work of another and as such should reflect the vision of the other not the peer-reviewer.
I differ to the editors of the Journal on this language use issue as the "journal" is their creation and it is the senior editors prerogative to pursue his/her vision of the journal.
In my overall review I recognize 2 creative others and myself a non-creative observer.
Line No. | Text | Comment |
17 | quality of the meat produced | Possible ambiguity. Method of production has been portrayed in the large “quality” research project in the EU. So when we use the word “quality” to mean the organoleptic valuation of the eating experience we have to make that clear. If the animal suffered in transit even though undetectable post slaughter, the EU definition would still rate that product as having a quality deficit. |
66, 83, 86 | implications on meat quality | implications on organoleptic measures of meat quality? |
94 | Image of the 2000 + Mob | How is it possible to maintain contact with the sheep at the head of this bunch, which appears to be 500m at least away from the drivers? Some mention of the temperament of the sheep may be helpful. Do they bunch by nature making livestock protection dogs useful? Are they easily scattered? What is their flight distance? In this photograph what is making those sheep 500 m away move away from the drivers? Are we way-over penetrating the flight zone of those sheep close to the drivers. Remarkable photograph! |
225 | 4h is 0.8 m2 /100 kg lamb | I doubt that lambs weigh 100kg so somewhere in this document there should be an indication that export market lamb meat is derived primarily from 45kg live weight ? lambs or whatever the truth is. |
314 | Hability spelling | ability |
I would appreciate a description of the dogs. In one of the photos, Figure 2, there is a mixed breed dog; tail carriage and size is incompatible with a border collie and it may be a tri-colour I cant be sure from the photo. In Canada we have 3 types of dog associated with sheep management generally based on behaviour which is breed determined;
1. Bring the sheep to me dogs - Border collies
2. Drive the sheep away from me - Australian Blue Heelers
3. Stay with the sheep and don't come when your are called as, you dog, don't have a name - Large white central European Breeds.
This area of animal welfare and handling may be absent from this review as the research has not been done. This paper did raise this in my mind as a future area of research in the Chilean context.

Author Response
quality of the meat produced | Possible ambiguity. Method of production has been portrayed in the large “quality” research project in the EU. So when we use the word “quality” to mean the organoleptic valuation of the eating experience we have to make that clear. If the animal suffered in transit even though undetectable post slaughter, the EU definition would still rate that product as having a quality deficit. According to Warriss (2000) meat quality is composed of several concepts including sanitary, organoleptic, quantitative (weight and tissue composition of carcass), technological aspects such as pH, WHC and colour, and also ethical quality (production and handling). In this study we did not use organoleptic measures…we used technological measures with instruments. Therefore we prefer to stay with the broader term of meat quality in general, in no case organoleptic. We mainly refer to quantitative aspects like carcass yield and trimmings due to bruises, to pH and colour measured instrumentally, and of course ethical quality . Warriss, P. 2000. Meat Science: An Introductory Text. CABI Plublishing, Oxon, UK. |
implications on meat quality | implications on organoleptic measures of meat quality? As explained above, pH and colour were measured with equipment and therefore we consider them instrumental (technological) measures of meat quality. Bruises were appraised visually, but they are not usually considered within organoleptic measures of meat quality….. |
Image of the 2000 + Mob | How is it possible to maintain contact with the sheep at the head of this bunch, which appears to be 500m at least away from the drivers? Some mention of the temperament of the sheep may be helpful. Do they bunch by nature making livestock protection dogs useful? Are they easily scattered? What is their flight distance? In this photograph what is making those sheep 500 m away move away from the drivers? Are we way-over penetrating the flight zone of those sheep close to the drivers. Remarkable photograph! Thank you! We have added further explanation in line 93-97: Basically, several people on horseback and their dogs distribute themselves on the fields and start rounding up the sheep; as the sheep are bunched one driver will lead in the front showing the way towards the corrals, whilst the others remain around the back and sides of the herd. The procedure can take several hours depending on the size of the farms. |
Hability spelling | Ability corrected line 340 now |
I would appreciate a description of the dogs. In one of the photos, Figure 2, there is a mixed breed dog; tail carriage and size is incompatible with a border collie and it may be a tri-colour I cant be sure from the photo. In Canada we have 3 types of dog associated with sheep management generally based on behaviour which is breed determined;
Thank you! This was an interesting comment we had not thought about including!. I found two articles, unfortunately in Spanish but with English summary, that I would like to share with reviewer 1 and have added 2 sentences about the dogs characteristics in lines 137-142
Magellan sheepdogs are the result of breeding between British sheepdogs arrived in Chile in the late nineteenth and early twentieth centuries and have developed a high adaptation to the Chilean Patagonia climate conditions (Tafra et al 2014). They have some similarities in coat colour and markings with Bearded Collies, Border Collies and Bobtails, however, the colour tones within and between the coat colour patterns seen in Magellan sheepdogs show distinct features from other traditional Collie dog breeds (Barrios et al 2016).

Reviewer 2 Report
The manuscript is a comprehensive review of the work done on lamb
transport by the authors over a long-term period. A careful review of
language used, along with grammar should be conducted, although most of the manuscript
reads well. The formatting may not be final, but the left-hand column in
Table 1 should be redone to read more easily. As the authors reference
the 2013 regulations put into place in Chile, it would be helpful to
include a more thorough description of what is included in the
regulation, how enforcement is structured, what challenges to the
implementation exist and what type of training is available to meet the
standards, besides driver training. Is there a program in place to
educate farmers and handlers? It would also be beneficial if the authors
included a concluding section in which they could perhaps identify
potential solutions to challenges they have identified.Overall, this is
interesting work.
Author Response
Reviewer 2
The manuscript is a comprehensive review of the work done on lamb transport by the authors over a long-term period. A careful review of language used, along with grammar should be conducted, although most of the manuscript reads well.
The English language has been reviewed by a native English speaker.
The formatting may not be final, but the left-hand column in Table 1 should be redone to read more easily.
We tried to format the tables again, although we assume that the paper will be formatted by the editors and will be sent to us for galley proof (whenever possible, tables should be formatted horizontally to fit, or otherwise size of font reduced)
As the authors reference the 2013 regulations put into place in Chile, it would be helpful to include a more thorough description of what is included in the regulation, how enforcement is structured, what challenges to the implementation exist and what type of training is available to meet the standards, besides driver training. Is there a program in place to educate farmers and handlers?
The following information was added in lines 153-167:
The Chilean regulations for the protection of animals cover livestock in industrial production systems (Decreto 29, Chile), livestock transport (Decreto 30, Chile) and slaughter (Decreto 28, Chile); the general aim in all cases is to acknowledge that farm animals are sentient beings and that there is a need to reduce pain and unnecessary suffering in them. The regulation for industrial production systems (Decreto 29) includes indications regarding feeding, health and housing conditions; there are specific prohibitions during handling like hitting, pricking or grabbing animals from different parts of the body and there are training obligations. These make it compulsory to have at least one person trained as animal handler in each productive system, which could be the farmer himself or a designated person with a training certificate. The training consists of a 16 h course given by educational institutions that are recognized for this purpose by the competent authority (the Agriculture and Livestock Service (SAG) of the Chilean Ministry of Agriculture). Enforcement is structured in the way of audits by SAG, but is still at an early stage at farm level and more effort has been put on the enforcement to livestock transporters and slaughterhouses. However efforts are being made by producer associations, educational institutions and SAG to develop and spread information on good handling practices in order to improve animal welfare and create awareness on the legislation.
It would also be beneficial if the authors included a concluding section in which they could perhaps identify potential solutions to challenges they have identified. Overall, this is interesting work.
We have added a conclusion chapter with potential solutions for the problems shown see line 547:
Rounding up and driving suckling lambs from distant fields in extensive sheep farming systems, like the Chilean Patagonia, is stressful and reduces energy reserves of the animals; a possible solution to reduce stress before loading in these cases could be rounding up the animals some days before the expected transport, leaving them close to the loading point.
Although facilities for the loading and transport of lambs are appropriate in the Chilean Patagonia, deficiencies were observed at the level of small producers particularly in the central-southern region; creating awareness to farmers and transporters about animal welfare, appropriate infrastructure and developing good handling practices at this level should receive assistance by governmental institutions.
Misshandling the lambs, as it was often observed when using driving aids incorrectly, grabbing them from different parts of the body or pulling them from the wool, not only has consequences on the welfare of animals but also on product quality in Chile, as determined by the high percentage of bruised carcasses. Further enforcement of the legislation for the protection of animals in industrial sheep farming systems and during transport should lead to improvements in this respect.
The duration of lamb transport in Chile is often increased due to geographical reasons and bad roads; the space allowances used for lambs are similar in long and short distance transport. The dehydration and increased concentration of betahydroxybutyrate in the lambs after transport of 24 h confirm that lambs should be fed and watered during these journeys. More emphasis on enforcement of the law is needed to improve the welfare and reduce losses in terms of mortalities, live and carcass weight, as well as meat quality of lambs.
Efforts should be made to reduce the time lambs have to remain in lairage at the slaughterhouses in order to improve animal welfare and meat quality. Although the relationship between muscle glycogen concentration and pH in lambs seems less clear that in cattle, within the practical recommendations to address high pH problems in Chile are reduction of exercise and stress to the minimum possible on farm before transport, and reducing transport and lairage times. This stops lambs from using their energy reserves before being slaughtered and it mainly requires efforts in better planning of the timing of transport and slaughter operations. Meat pH measurements and bruising should be used at slaughter plants as indicators of meat quality, and also of animal welfare.